# Molecular Epidemiologic and Geo-Spatial Characterization of *Staphylococcus aureus* Cultured from Skin and Soft Tissue Infections from United States-Born and Immigrant Patients Living in New York City

**DOI:** 10.3390/antibiotics12101541

**Published:** 2023-10-14

**Authors:** Lilly Cheng Immergluck, Xiting Lin, Ruijin Geng, Mike Edelson, Fatima Ali, Chaohua Li, TJ Lin, Chamanara Khalida, Nancy Piper-Jenks, Maria Pardos de la Gandara, Herminia de Lencastre, Alexander Tomasz, Teresa H. Evering, Rhonda G. Kost, Roger Vaughan, Jonathan N. Tobin

**Affiliations:** 1Department of Pediatrics, University of Chicago, Chicago, IL 60637, USA; limmergluck1@uchicago.edu; 2Morehouse School of Medicine, Atlanta, GA 30310, USA; xlin@msm.edu (X.L.); rgeng@msm.edu (R.G.); faali@msm.edu (F.A.); cli@msm.edu (C.L.); 3Interdev, Roswell, GA 30076, USA; medelson2000@yahoo.com; 4Clinical Directors Network (CDN), New York, NY 10018, USA; tlin@cdnetwork.org (T.L.); ckhalida@cdnetwork.org (C.K.); njenks@sunriver.org (N.P.-J.); 5Institut Pasteur, Université Paris Cité, Unité des Bactéries Pathogènes Entériques, Centre National de Référence des Escherichia coli, Shigella et Salmonella, F-75015 Paris, France; maria.pardos-de-la-gandara@pasteur.fr; 6Center for Clinical and Translational Science, The Rockefeller University, New York, NY 10065, USA; lencash@mail.rockefeller.edu (H.d.L.); alexander.tomasz@rockefeller.edu (A.T.); rhonda.kost@rockefeller.edu (R.G.K.); rvaughan@rockefeller.edu (R.V.); 7Laboratory of Molecular Genetics, Instituto de Tecnologia Química e Biológica António Xavier (ITQB NOVA), 2780-157 Oeiras, Portugal; 8Division of Infectious Diseases, Department of Medicine, Weill Cornell Medicine, New York, NY 10065, USA; evering@med.cornell.edu

**Keywords:** skin and soft tissue infections (SSTIs), antibiotic resistance, antimicrobial resistance, staph aureus, methicillin-resistant *Staphylococcus aureus* (MRSA), methicillin-sensitive *Staphylococcus aureus* (MSSA), immigrant health, Federally Qualified Health Centers (FQHCs), practice-based research networks (PBRNs)

## Abstract

(1) Background: With increasing international travel and mass population displacement due to war, famine, climate change, and immigration, pathogens, such as *Staphylococcus aureus* (*S. aureus*), can also spread across borders. Methicillin-resistant *S. aureus* (MRSA) most commonly causes skin and soft tissue infections (SSTIs), as well as more invasive infections. One clonal strain, *S. aureus* USA300, originating in the United States, has spread worldwide. We hypothesized that *S. aureus* USA300 would still be the leading clonal strain among US-born compared to non-US-born residents, even though risk factors for SSTIs may be similar in these two populations (2) Methods: In this study, 421 participants presenting with SSTIs were enrolled from six community health centers (CHCs) in New York City. The prevalence, risk factors, and molecular characteristics for MRSA and specifically clonal strain USA300 were examined in relation to the patients’ self-identified country of birth. (3) Results: Patients born in the US were more likely to have *S. aureus* SSTIs identified as MRSA USA300. While being male and sharing hygiene products with others were also significant risks for MRSA SSTI, we found exposure to animals, such as owning a pet or working at an animal facility, was specifically associated with risk for SSTIs caused by MRSA USA300. Latin American USA300 variant (LV USA300) was most common in participants born in Latin America. Spatial analysis showed that MRSA USA300 SSTI cases were more clustered together compared to other clonal types either from MRSA or methicillin-sensitive *S. aureus* (MSSA) SSTI cases. (4) Conclusions: Immigrants with *S. aureus* infections have unique risk factors and *S. aureus* molecular characteristics that may differ from US-born patients. Hence, it is important to identify birthplace in MRSA surveillance and monitoring. Spatial analysis may also capture additional information for surveillance that other methods do not.

## 1. Introduction

There has been a steady rise in the movement of people across national borders via being tourists, university students, refugees, and migrants [1]. With this movement, there can also be a spread of novel pathogens across geographic regions. One such pathogen, *Staphylococcus aureus* (*S. aureus*), is commonly found on the skin. *S. aureus* can lead to a variety of clinically significant invasive and non-invasive infections, e.g., skin and soft tissue infections (SSTIs), osteomyelitis, pneumonia, and bloodstream infections. Methicillin-resistant *S. aureus* (MRSA) infections have been associated with serious diseases, resistance to major classes of antibiotics, and increased morbidity and mortality [2,3]. MRSA was previously only reported in hospitals and other health care settings (HA-MRSA); however, in the late 1990s and early 2000s, MRSA infections had sharply increased at alarming rates in community settings among people with no established risk factors (e.g., occupational exposure, residence in long-term care facilities, history of multiple hospitalizations or surgeries, and chronic medical conditions, such as renal disease, diabetes, and hypertension) [4,5,6,7]. These community-associated infections due to MRSA (CA-MRSA) accounted for the majority of purulent SSTIs in outpatient settings, most often presenting in the form of abscesses and cellulites [8,9,10].

In the United States (US), the most common CA-MRSA clone is USA300. This clone, originating in the US, was first reported in 1999 among prisoners, athletes, and children [11]. Beginning in 2000, in the US, MRSA USA300 was recognized as not only replacing other *S. aureus* strains but also adding to the overall burden of *S. aureus*-related diseases [3,11,12]. It has since spread to other continents, including South America, Europe, and Africa [11,13,14,15]. Prior research has shown that international travel has contributed to the worldwide spread of MRSA USA300 [9,16]. Although there has been much research investigating SSTIs due to *S. aureus* in travelers, less is known about *S. aureus* infections among immigrants and refugees, and whether the duration of time residing in either country of birth or country of residence impacts risk for *S. aureus* colonization or infection. Specifically, little is known about the prevalence of MRSA USA300 in immigrant populations in the US and the variables that play a role in *S. aureus* USA300 infections within these populations.

*S. aureus* USA300 is the causal pathogen for the majority of SSTIs, and specific virulence factors tend to also occur in this genotype. For example, arginine catabolic mobile element (ACME), a virulence factor encoded by the gene *arcA*, is usually present in USA300 along with Panton Valentine Leucocidin (PVL). However, in South America, it was found that some clinical isolates were genetically similar to USA300 but lacked ACME. These isolates were designated USA300 Latin American variant (USA300-LV) [17].

For many immigrants who lack health insurance, Community Health Centers (CHCs) are an accessible ‘medical home’ for acute and preventive care. CHCs provide preventive and primary care services to medically underserved communities, including immigrant, refugee, and migrant populations [18]. 

Our study examines the prevalence of MRSA and MSSA USA300 SSTI among non-US-born (immigrant) compared to US-born participants who sought treatment for SSTIs from CHCs in New York City and explores the geo-spatial distribution in relation to the molecular epidemiologic characteristics. We hypothesize an increased risk for MRSA USA300 SSTI among immigrant populations, which correlates with the duration of time spent in the US. In our model, we include demographic and environmental exposure risk factors and explore the geo-spatial relatedness of ethnocentric communities with the development of *S. aureus* USA300 SSTI. This is a follow-up study to two prior studies by this team that reported on the prevalence of CA-MRSA and CA-MSSA among US- vs. non-US-born populations [19,20].

## 2. Results

### 2.1. Overview of Enrollment Population

The clinical trial and population screened and enrolled have been described previously [20]. Briefly, 602 patients with SSTIs were approached to enroll in the trial between 2015 and 2017 from six New York City CHCs and hospital emergency departments (EDs), and 181 patients were excluded; the main reasons for exclusion included that 141 declined to participate in the study, and 13 fell outside of the age requirement or were using antibiotic treatment at enrollment. A total of 421 patients with SSTIs were enrolled in this study. Figure 1 shows the enrollment scheme, including reasons for exclusion. An additional 235 patients were excluded because no *S. aureus* was isolated from the wound. In total, 46 of 186 (24.7%) responded that their place of birth was outside of the US. The majority (29 of 46, 63%) of the participants born outside of the US were born in Latin American countries (Figure 2).

### 2.2. Concordance between S. aureus Wound and Carriage

SSTI wound cultures demonstrated *S. aureus* from 186 (44.2%) participants. Of the 186 participants with a confirmed *S. aureus* infection, 63.4% (118) also showed evidence of *S. aureus* carriage (positive *S. aureus* growth from a swab taken from non-infected skin or mucous membrane at the time of enrollment, including nasal, axilla, and groin areas). The concordance between wound SSTI and carriage sites for methicillin sensitive isolates was 91.5% (108 out of 118) and 85% (84 out of 99) concordance between wound SSTI and carriage sites for *S. aureus* protein A typing (*spa* typing) via polymerase chain reaction. 

### 2.3. Comparison between US-Born and Non-US-Born

US-born and non-US-born patients enrolled in the study were different socio-demographically (Table 1). Descriptions of variables can be found in Appendix A. A higher proportion of US-born patients were black (45.8%; *p* = 0.008), elderly, or pediatric (12.3% and 6.2%, respectively; *p* = 0.013), single (not married, 80.2%; *p* < 0.0001), had a high school education or less (67.5%; *p* = 0.007), and had public health insurance (72.8%; *p* < 0.0001) compared to non-US-born patients. A higher proportion of non-US-born patients were Hispanic (82.5%; *p* = 0.007), married (63.6%; *p* < 0.0001), had a bachelor’s degree or higher (26.7%; *p* = 0.0007), and had neither public nor private health insurance (45.5%; *p* < 0.0001). 

There was a trend towards significant differences (*p* = 0.11) between the distribution of a type of *S. aureus* isolates among US-born compared to those isolates from non-US-born patients: 58% MRSA among US-born compared to 43.5% among non-US-born (Table 2). However, a significant difference in genotype was observed, where 46.9% of US-born patients had USA300 strain, compared to 28.3% of non-US-born patients (*p* = 0.04). Among the virulent factors tested, only a statistically significant difference was seen for ACME type I or III (*p* = 0.04) between these two populations.

Combinations of *S. aureus* stratified by methicillin sensitivity and genotype (i.e., MRSA USA300, MRSA non-USA300, MSSA USA300, and MSSA non-USA300) (Table 3) did not reveal any significant differences between those born in the US compared to those born outside of the US for all categories. Among non-US-born, MSSA USA300 was not seen nearly as frequently as MSSA non-USA300.

### 2.4. Risk Factors for S. aureus MRSA SSTI and USA300 SSTI by Birthplace

Overall, demographics were similar between patients with MRSA and MSSA infections (Appendix A). However, their social interactions and environment did reveal some statistically significant differences: contact crowding was significantly different between MRSA and MSSA (*p* = 0.03), and contact with animals (e.g., owning pets or occupation involving animals) trended towards significance (*p* = 0.08). A higher proportion of patients with MSSA infections reported sharing personal hygiene products (65.4%) compared to 47.3% patients with MRSA infections (*p* = 0.06). Differences were observed for molecular profiles between MRSA and MSSA: 58.2% of MRSA isolates were USA300 genotype, whereas 20.0% of MSSA were USA300 genotype. Additionally, a higher proportion of MRSA isolates contained the following genes: mecA, ACME, and PVL. 

The demographics between patients with SSTIs from USA300 *S. aureus* and non-USA300 were not statistically different (Appendix A) except for birthplace, comparing patients born in and outside of the US (*p* = 0.04). Contact with animals was significantly different between USA300 and non-USA300 (45.8% and 27.0%, respectively; *p* = 0.03). The majority of USA300 *S. aureus* isolated from wound cultures contained PVL (82%; *p* < 0.0001), mecA (78%; *p* < 0.0001), and ACME (type I and III; 72%; *p* < 0.0001) genes.

Appendix A compares US-born and non-US-born patients for MRSA and MSSA. Within MRSA, a higher proportion of US-born patients are black (45.5%, *p* = 0.009), single (76.6%, *p* = 0.001), and have public health insurance (70.2%, *p* = 0.001). A higher proportion of non-US-born patients are Hispanic (84.2%, *p* = 0.06), had a bachelor’s degree or above (36.8%, *p* = 0.003), and had no health insurance (47.4%, *p* = 0.001). Within MSSA, a higher proportion of US-born patients were single (85.3%, *p* = 0.0003) and had public health insurance (76.5%, *p* = 0.002). A higher proportion of non-US-born patients were married (60.0%, *p* = 0.0003) and had no health insurance (44.0%, *p* = 0.002). There was no significant difference in socio-environmental factors between those with MRSA and MSSA in their wound cultures.

Appendix A compares US-born and non-US-born patients by genotype (USA300, non-USA300). A higher proportion of US-born patients with USA300 were black (50.0%, *p* = 0.03), had some college education (31.6%, *p* = 0.03), and had public healthcare insurance (68.4%, *p* = 0.0004). A higher proportion of non-US-born patients with USA300 were married (66.7%, *p* = 0.01) and had no health insurance (58.3%, *p* = 0.0004). Within non-USA300, a higher proportion of non-US-born patients were Hispanic (85.2%, *p* = 0.009), married (62.5%, *p* = 0.0001), had a bachelor’s degree or higher (27.3%, *p* = 0.034), and had no health insurance (40.6%, *p* = 0.002). US-born patients had a higher percentage who were children or over 65 years old (16.3% and 7.0%, respectively, *p* = 0.022), single (81.4%, *p* = 0.0001), and had public health insurance (76.7%, *p* = 0.002). There was no significant difference in other socio-environmental factors between MRSA and MSSA.

### 2.5. Molecular Features of S. aureus MRSA SSTI and USA300 SSTI by Birthplace and Characteristics of Current Residence

Within each category of birthplace, the four combinations of methicillin sensitivity stratified by USA300 and non-USA300 genotype were compared (Table 4). All four molecular features (PVL, *mecA*, ACME, and SCC*mec*) were significantly different among US-born patients: the majority of MRSA USA300 had PVL (90.0%, *p* < 0.0001), *mecA* (100%, *p* < 0.0001), ACME Type I (86.7%, *p* < 0.0001), and SCC*mec* type IVa (96.7%, *p* < 0.0001). In the non-US-born patients, there was no significant difference in PVL across the four categories of methicillin sensitivity. However, the majority of MRSA USA300 had *mecA* (100%, *p* < 0.0001), ACME (77.8%, *p* = <0.0001), and SCC*mec* type IVa (100%, *p* < 0.0001). There was no significant difference in the molecular characteristics with regard to specific genes (*mec*A, SCC, ACME, and PVL) when comparing US-born and non-US-born patients for methicillin sensitivity (Appendix A). There was also a non-significant difference in molecular characteristics when comparing US-born and non-US-born patients for genotype (Appendix A). Specifically, although the distribution of ACME was not significantly different in USA300 between US-born and non-US-born patients, 6 of the 13 non-US-born patients with USA300 (46.2%) had the Latin American variant (USA300-LV; defined as USA300 genotype but negative for ACME), and 8 of the 37 US-born patients (21.6%) with USA300 had USA300-LV. Of the eight US-born patients with USA300-LV, six of them identified as Hispanic (75%). In total, ten patients with USA300-LV lived in Hispanic ethnocentric neighborhoods (five non-US-born and five US-born patients). Other ethnocentric neighborhoods include white (two in total, one US-born and one non-US-born) and black (two US-born only). Most commonly, eleven patients were in “High-Rise Renters” Environmental Systems Research Institute, Inc. (ESRI)-defined neighborhood segments (seven US-born and four non-US-born). One patient each lived in “Downtown Melting Pot”, “International Marketplace”, and “City Strivers”. 

### 2.6. Risk for S. aureus USA300 Skin and Soft Tissue Infection Based on Immigration Status (US-Born vs. Non-US-Born)

A logistic regression was performed to determine the risk for USA300 (Table 5). In the unadjusted model, only contact with animals was statistically significant, where patients who owned pets or worked with animals were 2.5 times more likely to have USA300 infections than non-USA300. After adjusting the model for birthplace, crowded living, contact animal, and age, people born in the US were 3.2 times more likely to have USA300 than non-USA300 compared to non-US-born patients. Since birthplace and time spent in the US were correlated, only birthplace was included in the final model. 

### 2.7. Geo-Spatial Distribution of Patients with S. aureus Skin and Soft Tissue Infections

In total, 186 unique patient residential locations were georeferenced, reflecting 94 CA-MRSA and 92 CA-MSSA. We identified 59 patients who had the *S. aureus* USA300 genotype compared to 99 patients who had non-USA300; all point locations were mapped to the respective ethnocentric NYC map. Figure 3 shows the spatial distribution of patients infected with MRSA and MSSA with the boundaries of specific race/ethnic neighborhoods outlined [21], using data from the American Community Survey (ACS); the base layer map indicates areas where residents were the majority of a specific US race (white, black) and ethnicity (Hispanic). For areas for which the majority of the population was immigrants, categorization was based on continent of birth origin (Asia, Europe, Latin America) or region (Caribbean). Among non-US-born participants, 87% (40 of 46) resided in three boroughs: Manhattan (20%), the Bronx (37%), and Brooklyn (30%). In comparison, among US-born patients, 95% (77 of 81) lived in these three boroughs: Manhattan (32%), the Bronx (14%), and Brooklyn (31%). US-born and non-US-born patients mostly lived in the same racial/ethnic majority neighborhood as the race/ethnicity in which they self-identified (Appendix A). For example, patients who identified as Hispanic mostly lived in Hispanic neighborhoods for both US- and non-US-born patients at 63.4% and 59.4%, respectively. We identified that the majority of participants, regardless of country of birth, resided in neighborhoods classified as ‘High-Rise Renters’, using ESRI’s Tapestry neighborhood assignment (Appendix A) [22,23].

### 2.8. Hot/Cold Spot Analyses for S. aureus and Genotype-USA300

We found no clusters of MRSA SSTIs nor clusters of MSSA SSTIs. However, when we stratified by genotype (USA300 vs. non-USA300), we were able to ascertain clustering of *S. aureus* USA300 in three areas (Figure 4). The hotspot pattern for the non-USA300 genotype differed and covered a wider area over three boroughs (Manhattan, Bronx, and Brooklyn). Two of these hotspots (Manhattan and Bronx) are closely located to two CHCs. Another cluster in Brooklyn is in a neighborhood which has many residents from eastern Europe. In Brooklyn, two hotspots are in neighborhoods where residents were more than 50% non-US-born; we found no evidence of cold spots in the data (four non-USA300 patients lived outside of the New York City limits and were therefore not included in the hot/cold spot analysis).

## 3. Discussion

In this study, we examined several individual and neighborhood-level socio-demographic and geo-spatial co-variates in relation to the molecular characteristics of *S. aureus* type (MRSA, MSSA) and genotype (USA300, non-USA300) in patients with SSTIs from six CHCs with an examination of birthplace (US-born, non-US-born) and duration in place of birth, compared to where they developed their *S. aureus* infection. Previously, Jenks et al. reported that SSTIs caused by MSSA were more likely to occur among non-US-born participants [19]. This paper explores the dynamics around place (location of participants’ self-reported residences at the time of their *S. aureus* SSTI) and specifically addressed the impact of the number of years living in the US on the type of *S. aureus*, including the presence or absence of a mobile genetic element, ACME. Explanations for why more MSSA was seen in immigrants compared to those born in the US may be related to birth country antibiotic usage, e.g., other countries outside of the US may not have access to or consume as many antibiotics as those living in the US; therefore, bacterial selective evolutionary pressures may reduce the risk for MRSA infections. In a study conducted in Norway, it was found that immigration may be contributing to increasing community onset-MRSA infections [24]. Other factors that may explain this phenomenon may be related to why in Jenks’ study, the non-US-born patients were presenting as a first-time infection compared to the US-born patients, who were more likely to have presented with prior SSTIs. Hagmann et al. postulated that this may be related to simply exposure to *S. aureus* within households that serve as a reservoir, further emphasizing the importance of exploring the relationship between spatial parameters, household, and neighborhood characteristics and their role in risk for *S. aureus* SSTIs [25].

Overall, non-US-born patients enrolled in this study were more likely to have a higher education (i.e., bachelor’s degree or above), have no health insurance, and be married, whereas US-born patients were more likely to have a high school degree or lower educational attainment, public health insurance, and be single. This emphasizes some of the unique challenges non-US-born people face in the US with accessing healthcare. Potential barriers include little or no English language fluency or literacy and a lack of insurance coverage. Using the ESRI neighborhood characterization, we also found that regardless of place of birth, most of the participants lived in settings very characteristic of NYC, and no significant differences were seen in the types of ESRI neighborhoods between those with MRSA born in the US compared to those not born in the US. 

Overall, we only found a trend towards significance (*p* = 0.11) where US-born patients were more likely to be infected with MRSA compared to non-US-born patients. While we saw no place-based factors that differed significantly between US-born and non-US-born patients that could explain this trend, it is suggested that infections due to *S. aureus* may be related to conditions very typical of an urban, dense environment, including multiunit high-rise housing, multi-generational, crowded households, and diverse populations. The ESRI neighborhood assignment, ‘High-Rise Renters’, accounts for the majority of those with SSTIs for both US-born and non-US-born patients. This composite profile provides insights into place-based factors which may be risk factors for *S. aureus* infections, regardless of the race/ethnic makeup of a neighborhood; fundamentally, communities that are very typical of those described as ‘High-Rise Renters’ include culturally diverse neighborhoods.

We found that frequent exposure to animals, such as owning a pet or working in an animal facility, was a risk factor for a USA300 SSTI. Other studies have shown that animals, such as household pets and farm animals, carry and transmit *S. aureus* strains similar to humans [26,27,28]. Since companion animals are closely integrated with human environments, they may be picking up human *S. aureus* strains that commonly circulate in their environment. For example, in the US, CC8, for which USA300 has been found in dogs and cats [29,30,31,32]. This increase in USA300 among domestic animals not seen with other clonal strains may be a reflection of the predominant prevalence of USA300 carriage among humans. The number of years a patient has been living in the US did not affect genotype. The majority (71%) of USA300 were MRSA. Additionally, the majority (71%) of USA300 were from participants born in the US; unsurprisingly, the Latin American USA300 variant (LV USA300) was more common in participants born in Latin America. Some molecular characteristics of LV USA300 include lacking ACME and carrying SCCmecIVc [13]. The number of years a non-US-born patient has been living in the USwas not significantly different between non-US-born patients with a USA300 SSTI compared to a non-USA300 SSTI. This may suggest that birthplace may be a stronger contributing factor for the genotype of a *S. aureus* SSTI than the current geographic location of residence and environment. However, Kadariya et al. investigated *S. aureus* nasal carriage from two geographic populations, Bhutanese refugees living in Nepal and Bhutanese refugees living in Ohio, and found that the most common *spa* types were different in the two populations [33], suggesting a stronger impact on carriage is tied to current environmental exposures.

The two strongest predictors for a USA300 SSTI by logistic regression were birthplace and *S. aureus* type (MRSA vs. MSSA). Being born in the US and infected with MRSA both increased the risk by three times for a USA300 SSTI. Patients’ birthplace affected the probability that they would be infected with *S. aureus* USA300. The skin microbiome, which is established early in life, may play a role in risk for USA300 SSTI later in life. Although the skin microbiome can differ between individuals, within an individual, it has been found to be relatively stable in a two-year timespan regardless of environmental changes [34]. This mechanism may differ from those who travel or immigrate, which contribute towards the transmission and or acquisition of new *S. aureus* strains [35].

A novel aspect of this research is the use of geo-spatial analysis, including assessing for clustering or hot spots stratified by *S. aureus* type Geo-spatial analysis was performed to examine visually where the SSTIs were occurring in relation to the ethnic neighborhoods in New York City. After mapping the locations, hot spot analysis was performed. From the hotspot analysis map, we can see that *S. aureus* USA300 are clustered together, suggesting that patients with this type of infection potentially share similar socioeconomic conditions and/or a common exposure source. However, when we compared patients with *S. aureus* USA300 to those with non-USA300 SSTIs, we found there was a statistically significant difference among those living in crowded housing. We found no statistically significant difference for the risk factors tied to socioeconomic conditions between people infected with *S. aureus* USA300 and *S. aureus* non-USA300 at the individual level. Most lived in a neighborhood with a racial/ethnic majority that was the same as their own identified race or ethnicity. This is suggestive that cultural norms and behaviors among the other considered characteristics may be similar for people living in the same ethnic communities. Communities with similar cultural behaviors have been demonstrated to also have similar strains causing *S. aureus* infections in other parts of the world [36,37,38,39,40]. Culturally sensitive and relevant interventions are important considerations in reducing health disparities [41], but it would be even more relevant to tailor these interventions in largely homogenous neighborhoods.

Limitations. Enrollment was limited to patients at six New York City CHCs and EDs, which provide primary healthcare to underserved communities, including many patients with Medicaid or with no health insurance. These patients may have sociodemographic and environmental factors unique to underserved populations and may not be representative of New York City or, more broadly, the US. Although information was collected on how long participants have been living in the US, one limitation is that no information on travel history or relatives visiting from outside of the US was gathered. Travel history may also contribute to exposure and susceptibility to MRSA or USA300 SSTIs. Lastly, the sample sizes were also too small to draw meaningful conclusions for some co-variates. Future studies with a more even distribution of US-born and non-US-born participants may also enhance the certainty of these findings.

Conclusion. With the increasing spread of antimicrobial resistance, including MRSA, and increased travel among residents within and between countries, it is important to include immigrant populations in MRSA surveillance and to assess birthplace and duration of residence in the US as part of healthcare. MRSA infections continue to impact our communities, both in the US and the rest of the world [39,40,42,43]. Combining individual risk factors with molecular epidemiologic and geo-spatial analyses adds another dimension for studying and monitoring the spread of MRSA and other organisms, potentially detecting clusters and outbreaks earlier, and implementing culturally sensitive interventions promptly, as well as understanding the underlying immunologic and microbiome mechanisms which may contribute to carriage and infection with *S. aureus*.

## 4. Materials and Methods

From 2015 to 2017, 602 patients who presented to six New York City (NYC) area Community Health Centers (CHCs) and community hospital emergency departments (EDs) with SSTIs were recruited to participate in the clinical trial. All patients had a clinical condition which met the definition for a *S. aureus* SSTI, using the clinical practice guidelines developed by the Infectious Disease Society of America (IDSA) [44]. Of 602 patients approached by participating clinicians to take part in the study, 421 patients consented and were enrolled, as previously described [19]. Inclusion criteria included: (1) age between 7 and 70 years; (2) fluency in English or Spanish; (3) self-reported affirmation to continue receiving care at the same CHC or ED for at least 12 months after time of enrollment; and (4) signs and symptoms of SSTI present at the time of recruitment. See Figure 1 for enrollment schema.

Study Setting. The six CHCs and EDs are part of Clinical Directors Network (CDN URL: www.CDNetwork.org), a primary care Practice-Based Research Network (PBRN) which collaborates with Federally Qualified Health Centers (FQHCs), community hospital EDs and other primary care safety-net practices, with The Rockefeller University Center for Clinical and Translational Science, and the Laboratory of Microbiology at the Rockefeller University for this study [19]. In this study, the CHCs and EDs are located in different boroughs of New York City and primarily serve the communities adjacent to their respective locations. Figure 5 shows the locations of the six FQHCs. Participants provided written informed consent in a language they understood (either English or Spanish), and the study procedures were approved by the respective Institutional Review Boards at CDN and The Rockefeller University. 

Study Design. This is a secondary data analysis of a previous prospective interventional study. The primary outcome is *S. aureus* SSTI, stratified by MRSA and methicillin-sensitive *S. aureus*, (MSSA); all *S. aureus* cultured from SSTIs were genotyped, stratified by USA300 and non-USA300. A comprehensive questionnaire [19,20] was developed by CDN and administered during home visits by a team of two community health workers/promotoras to patients who were diagnosed by their clinician as having an SSTI due to *S. aureus* on a culture sent to a local commercial clinical laboratory (BioReference). The survey questionnaire includes information on: demographics, co-morbidities scale, healthcare utilization scale, social network and environmental exposures, quality of life scale, and patient-centered outcomes survey. Appendix A details the survey questions pertaining to the demographic, social, and environmental conditions included in the analyses. There are two main predictors: the birthplace (US-born or non-US-born) and the time (years) immigrant participants have resided in the US. We analyzed the impact of years in the US in one-year increments and ascertained whether there were any statistically significant interactions between years in the US and outcomes of interest, MRSA/MSSA, or USA300/Non-USA300. Other co-variates included in our analyses involve participants’ demographic profile, including race, assigned sex at birth, social network, environmental exposures, and other variables previously reported [19]. We also aggregated and then re-categorized specific variables from a survey tool [20,45] which had themes, e.g., exposure to animals, personal hygiene, social network, crowded conditions, healthcare exposure, and type of wounds.

Molecular Characterization of *S aureus* isolates. *S. aureus* carriage was determined as previously described from the following areas: nasal, axilla, and groin area. A wound swab was also collected from the SSTI. Swabs were collected at the time of enrollment and sent to a commercial laboratory (BioReference Laboratories, Inc., Elmwood Park, NJ, US) to determine for evidence of *S. aureus* and then assessed for antibiotic phenotype across different antibiotic classes. Purified subcultures were then sent to the Laboratory of Microbiology (Tomasz Lab) at The Rockefeller University, where molecular characterization was performed as previously described and included: 1. *Spa* typing—based on the sequence of a polymorphic region of the *S. aureus*-specific gene encoding for the staphylococcal protein A (*spa*). 2. Multi-locus Sequence Typing (MLST)—based on the sequences of seven housekeeping genes of *S. aureus*. 3. Pulsed-field Gel Electrophoresis (PFGE), which identifies bacterial clones by partial digestion of their DNA and migration of the fragments generated on a gel by electrophoresis. 4. Molecular determination of the arginine catabolic mobile element (ACME) and the Panton–Valentine leukocidin (PVL) virulence factors were also performed, given these two genetic determinants are strongly associated with CA-MRSA. 5. Typing of the SCC*mec* gene cassette carrying *mecA* in all MRSA isolates by multiplex-PCR amplification. All molecular techniques were performed as previously described [39,40,41,42,43,44,45,46,47,48,49,50,51,52,53,54,55,56,57].

Statistical Analyses. A bivariate analysis using Chi Square test or Fisher Exact test was performed to assess differences between those born in the US and those identified as immigrant or non-US-born; differences in the prevalence of MRSA SSTI and MSSA SSTI, and then *S. aureus* USA300 and non-USA300, were ascertained between US-born and non-US-born. We also assessed the relationship between birthplace with specific individual and area level demographics, social and environmental exposures, and individual-level molecular data characterization. Univariate logistic regressions were conducted to obtain crude odds ratios (OR) of the primary outcome for each predictor, and 95% confidence intervals (95% CIs) were computed. Adjusted ORs (AOR) were obtained from a multivariate logistic regression model which contains specific predictors identified a priori. All analyses were carried out using two-tailed tests with an overall level of significance of 0.05 and were conducted with SAS (Version 9.4).

Geo-Spatial Analysis. We created an ethnocentric map of the five boroughs in New York City [21], which served as the base layer (study area), and then conducted spatial analysis based on *S. aureus* (MRSA, MSSA) and *S. aureus* genotype (USA300, non-USA300). Boundary files for the New York City NTA areas were downloaded from the *NYC OpenData* site [58]. The US Census boundaries were downloaded from the US Census Bureau Data webpage [59]. Patient data were received in spreadsheet format by Clinical Directors Network (CDN). Data included patient-level demographics, including point locations of place of residence at the time of enrollment and wound characteristics, and all molecular epidemiologic assays were performed in the Laboratory of Microbiology (Tomasz Lab) at The Rockefeller University.

The addresses (residential street number, name, and postal zip code) of the patients enrolled in the study were georeferenced. We assigned an ID for each enrolled patient to their respective US Census Block Group, Census Tract, and New York City Neighborhood Tabulation Area (NTA). The NTAs divide the city into small neighborhood planning areas used by New York City’s Planning Department to analyze population and demographic data. We also determined the point location for each of the 6 CHCs and aligned each patient with the respective CHC where enrollment occurred.

Characterization of neighborhoods was primarily conducted using two methods: (1) Using the New York Times on 21 January 2011, “Then, as Now—New York’s Shifting Ethnic Mosaic” as a guide, we created an ethnocentric map of the five Boroughs in New York City. The participant’s home address was plotted on a map of New York City and color-coded based on *S. aureus* (MRSA vs. MSSA) and *S. aureus* genotype (USA300 vs. nonUSA300). All mapping was conducted using ArcGIS Pro 3.0 (ESRI, Redlands, CA, USA) [60]. (2) We applied ESRI Tapestry Segmentation of US residential areas to each block group where a participant lived. ESRI developed a profile of US neighborhoods, detailing attributes of communities based on demographic and socioeconomic variables to identify unique consumer markets throughout the US. Included in this profile are 67 distinct market segments that then were summarized into 14 ‘LifeMode’ groups (focus is on demographic characteristics and consumer behavior patterns) and 6 urbanization groups (geographic and physical features of area). Full methodology for the assignment of these geographic spaces is detailed elsewhere [61]. 

Hot/Cold Spot Analyses. Next, we looked for hidden spatial patterns in the patient distribution using the Optimized Hot Spot Analysis (OHSA) tool (Esri ArcGIS Pro, Redlands, CA, USA). The OHSA tool enables us to analyze statistically significant patterns where clustering of cases (or controls) exists compared to the rest of the study area and is not likely due to random occurrence. We evaluated statistically significant clusters of (*S. aureus* USA300 and *S. aureus* non-USA300) populations. The OHSA requires an input feature. In our case, it is the participant’s location. The following parameters were used in the OHSA tool to perform the OHSA. The Incident Data Aggregation Method was set to count incidents within a hexagon grid; the study area was set to New York City’s boundaries; and the hexagon cell size was set to 1250 feet (this value is the height of the hexagon). Once executed, the OHSA tool determined the significance of a participant’s location within a hotspot by confidence intervals of 90%, 95%, or 99%. See Figure 3. The OHSA tool associates the participant’s location value with the neighborhood and then compares the neighborhood to the rest of the study area. If the value in a neighborhood is higher than in the rest of the study area, the participant is in a hot spot; if the value is much lower in a neighborhood, then the patient is deemed to be in a cold spot. We define ‘neighborhood area’ as a 1250 ft radius using a hexagon grid pattern around the patient’s residential address. The 1250 ft radius is roughly the mean size of a typical New York City block.

## Figures and Tables

**Figure 1 antibiotics-12-01541-f001:**
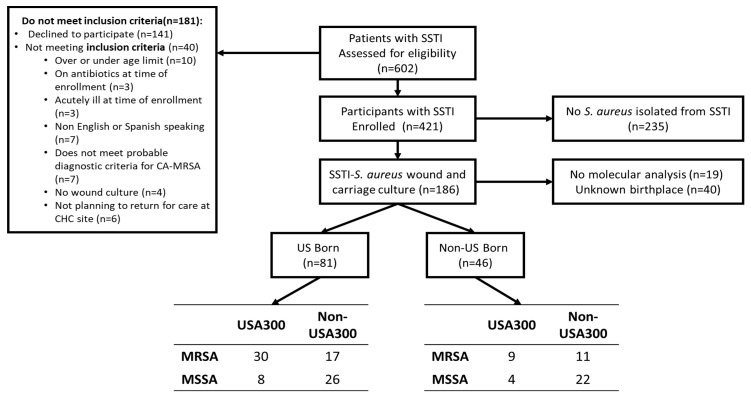
Enrollment scheme.

**Figure 2 antibiotics-12-01541-f002:**
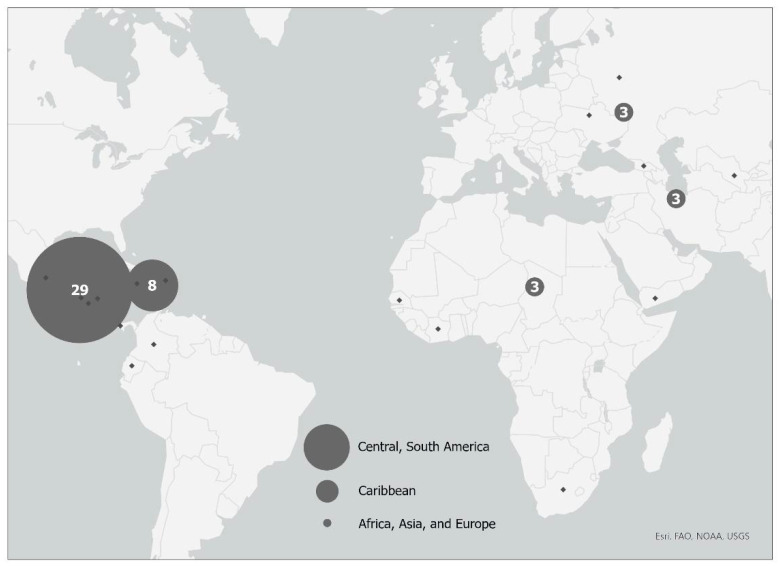
Global Map of non-US-born participants based on region and/or continent of birth. Circles mark the geographic location of the continents (Central America, South America, Africa, Asia, and Europe) or region (Caribbean) of origin for those participants born outside of the United States: size of gray circles correlates with relative number of participants from each continent or region.

**Figure 3 antibiotics-12-01541-f003:**
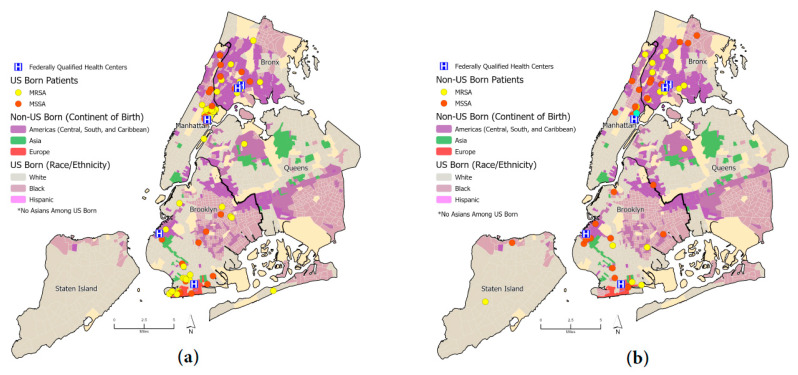
(**a**) Dot map of US-born patients stratified by methicillin sensitivity (MRSA, MSSA); (**b**) dot map of non-US-born patients stratified by methicillin sensitivity (MRSA, MSSA). Ethnic neighborhoods were obtained from the New York Times and the American Community Survey (ACS) [21].

**Figure 4 antibiotics-12-01541-f004:**
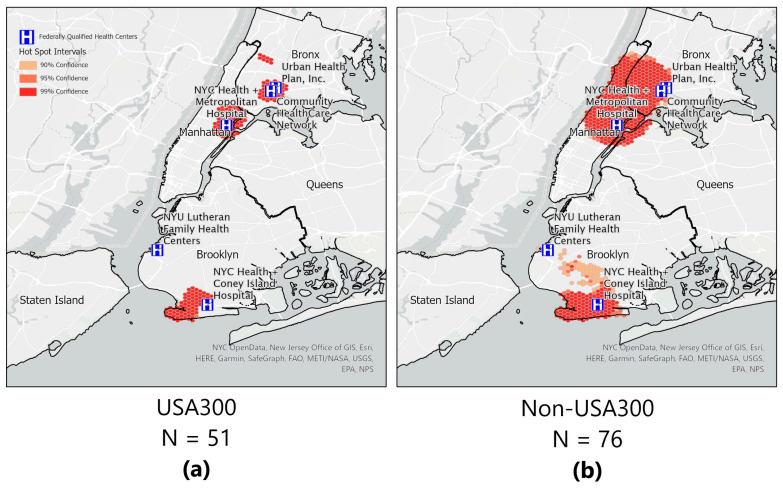
Optimized hotspot analyses. (**a**) Hot spots of areas with community-onset *Staphylococcus aureus* USA300 are shown, and (**b**) hot spots of areas with community-onset *Staphylococcus aureus* non-USA300 are shown. Hot spots of areas within the New York City boundary. Five of the six Community Health Centers are shown; one CHC was located outside the boundary. Confidence areas for the ‘hot spot’ are shown for 99% (darkest shade of orange); 95% (medium shade of orange); and 90% (lightest shade of orange).

**Figure 5 antibiotics-12-01541-f005:**
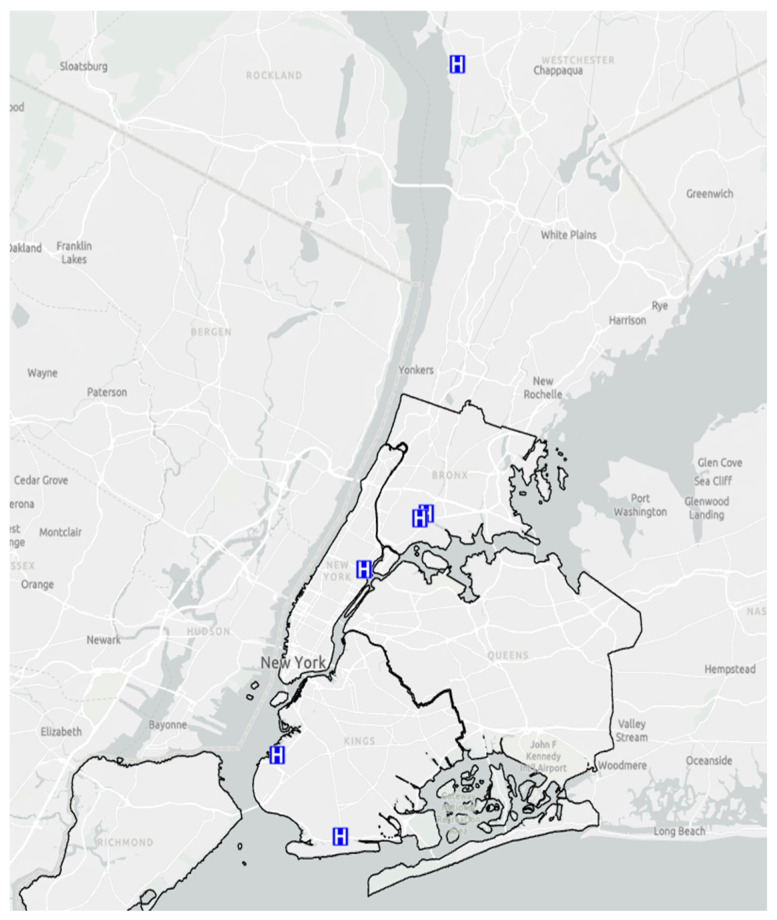
Locations of the Federally Qualified Health Centers (FQHCs), which are represented as “H” on a blue background.

**Table 1 antibiotics-12-01541-t001:** Population characteristics of US-born vs. non-US-born participants.

	US-Born (*n* = 81)	Non-US-Born (*n* = 46)	Total (*n* = 127)	*p*-Value
**Gender, *n* (%)**				0.75 ^1^
Female	34 (42.0%)	18 (39.1%)	52 (40.9%)	
Male	47 (58.0%)	28 (60.9%)	75 (59.1%)	
**Race, *n* (%)**				0.008 ^2^
Black	27 (45.8%)	3 (13.6%)	30 (37.0%)	
White	17 (28.8%)	6 (27.3%)	23 (28.4%)	
Other Race	15 (25.4%)	13 (59.1%)	28 (34.6%)	
Missing	22	24	46	
**Ethnicity, *n* (%)**				0.007 ^1^
Hispanic	43 (57.3%)	33 (82.5%)	76 (66.1%)	
Non-Hispanic	32 (42.7%)	7 (17.5%)	39 (33.9%)	
Missing	6	6	12	
**Age (years), *n* (%)**				0.01 ^2^
<19	10 (12.3%)	0 (0.0%)	10 (7.9%)	
19–45	43 (53.1%)	31 (67.4%)	74 (58.3%)	
45–65	23 (28.4%)	15 (32.6%)	38 (29.9%)	
>65	5 (6.2%)	0 (0.0%)	5 (3.9%)	
**Marital Status, *n* (%)**				<0.0001 ^1^
Couple	16 (19.8%)	28 (63.6%)	44 (35.2%)	
Single	65 (80.2%)	16 (36.4%)	81 (64.8%)	
Missing	0	2	2	
**Education, *n* (%)**				0.001 ^2^
High School or lower	54 (67.5%)	30 (66.7%)	84 (67.2%)	
College	21 (26.3%)	3 (6.7%)	24 (19.2%)	
Bachelor or higher	5 (6.3%)	12 (26.7%)	17 (13.6%)	
Missing	1	1	2	
**Health Insurance, *n* (%)**				<0.0001 ^1^
Private or Other	14 (17.3%)	10 (22.7%)	24 (19.2%)	
Public (Medicare or Medicaid)	59 (72.8%)	14 (31.8%)	73 (58.4%)	
None	8 (9.9%)	20 (45.5%)	28 (22.4%)	
Missing	0	2	2	
**Health Quality, *n* (%)**				0.77 ^2^
Good	50 (64.9%)	32 (71.1%)	82 (67.2%)	
Fair	22 (28.6%)	11 (24.4%)	33 (27.0%)	
Poor	5 (6.5%)	2 (4.4%)	7 (5.7%)	
Missing	4	1	5	
**Income, *n* (%)**				0.50 ^2^
<USD 40,000	56 (90.3%)	27 (84.4%)	83 (88.3%)	
≥USD 40,000	6 (9.7%)	5 (15.6%)	11 (11.7%)	
Missing	19	14	33	
**First Time Infection, *n* (%)**				0.22 ^1^
Yes	53 (69.7%)	36 (80.0%)	89 (73.6%)	
No	23 (30.3%)	9 (20.0%)	32 (26.4%)	
Missing	5	1	6	
**Crowding Life Environment, *n* (%)**				0.27 ^1^
Yes	47 (63.5%)	33 (73.3%)	80 (67.2%)	
No	27 (36.5%)	12 (26.7%)	39 (32.8%)	
Missing	7	1	8	
**Healthcare Exposure, *n* (%)**				0.24 ^1^
Yes	30 (39.5%)	13 (28.9%)	43 (35.5%)	
No	46 (60.5%)	32 (71.1%)	78 (64.5%)	
Missing	5	1	6	
**Animal Contact, *n* (%)**				0.17 ^1^
Yes	30 (39.0%)	12 (26.7%)	42 (34.4%)	
No	47 (61.0%)	33 (73.3%)	80 (65.6%)	
Missing	4	1	5	
**Had Wounds, *n* (%)**				0.15 ^1^
Yes	29 (37.2%)	11 (24.4%)	40 (32.5%)	
No	49 (62.8%)	34 (75.6%)	83 (67.5%)	
Missing	3	1	4	
**Social Network, *n* (%)**				0.76 ^1^
Yes	17 (22.4%)	9 (20.0%)	26 (21.5%)	
No	59 (77.6%)	36 (80.0%)	95 (78.5%)	
Missing	5	1	6	
**Household Crowding, *n* (%)**				0.11 ^1^
<2 People	25 (35.7%)	8 (21.1%)	33 (30.6%)	
>2 People	45 (64.3%)	30 (78.9%)	75 (69.4%)	
Missing	11	8	19	
**Personal Hygiene, *n* (%)**				0.35 ^1^
Not Sharing	28 (40.6%)	19 (50.0%)	47 (43.9%)	
Sharing	41 (59.4%)	19 (50.0%)	60 (56.1%)	
Missing	12	8	20	
**Hand Washing, *n* (%)**				0.53 ^1^
<10 Times/Day	53 (67.9%)	33 (73.3%)	86 (69.9%)	
>10 Times/Day	25 (32.1%)	12 (26.7%)	37 (30.1%)	
Missing	3	1	4	

^1^ Chi-Square *p*-value; ^2^ Fisher exact *p*-value.

**Table 2 antibiotics-12-01541-t002:** Molecular characteristics of *S. aureus* wound cultures stratified by birthplace.

	US-Born (*n* = 81)	Non-US-Born (*n* = 46)	Total (*n* = 127)	*p*-Value
***S. aureus*, *n* (%)**				0.11 ^1^
MRSA	47 (58.0%)	20 (43.5%)	67 (52.8%)	
MSSA	34 (42.0%)	26 (56.5%)	60 (47.2%)	
***S. aureus* Genotype, *n* (%)**				0.04 ^1^
USA300	38 (46.9%)	13 (28.3%)	51 (40.2%)	
Non-USA300	43 (53.1%)	33 (71.7%)	76 (59.8%)	
***S. aureus* + Genotype, *n* (%)**				0.17 ^1^
MRSA, USA300	30 (37.0%)	9 (19.6%)	39 (30.7%)	
MRSA, non-USA300	17 (21.0%)	11 (23.9%)	28 (22.0%)	
MSSA, USA300	8 (9.9%)	4 (8.7%)	12 (9.4%)	
MSSA, non-USA300	26 (32.1%)	22 (47.8%)	48 (37.8%)	
***mecA* Gene from *S. aureus*** **Wound, *n* (%)**				0.16 ^2^
Positive	42 (54.5%)	18 (39.1%)	60 (48.8%)	
Negative	32 (41.6%)	27 (58.7%)	59 (48.0%)	
Not Determined	3 (3.9%)	1 (2.2%)	4 (3.3%)	
Missing	4	0	4	
**ACME Gene from *S. aureus* Wound, *n* (%)**				0.04 ^2^
Negative	42 (54.5%)	36 (78.3%)	78 (63.4%)	
Type I	31 (40.3%)	9 (19.6%)	40 (32.5%)	
Type III	1 (1.3%)	0 (0.0%)	1 (0.8%)	
Not Determined	3 (3.9%)	1 (2.2%)	4 (3.3%)	
Missing	4	0	4	
**SCC*mec* Gene from *S. aureus* Wound, *n* (%)**				0.82 ^2^
Negative	1 (1.3%)	1 (2.2%)	2 (1.6%)	
IVa	35 (45.5%)	17 (37.0%)	52 (42.3%)	
IVb	1 (1.3%)	0 (0.0%)	1 (0.8%)	
IVc	2 (2.6%)	1 (2.2%)	3 (2.4%)	
IVg	1 (1.3%)	0 (0.0%)	1 (0.8%)	
IVh	1 (1.3%)	0 (0.0%)	1 (0.8%)	
Not Determined	34 (44.2%)	27 (58.7%)	61 (49.6%)	
Novel Type	2 (2.6%)	0 (0.0%)	2 (1.6%)	
Missing	4	0	4	
**PVL Gene from *S. aureus* Wound, *n* (%)**				0.36 ^1^
Positive	47 (61.0%)	23 (50.0%)	70 (56.9%)	
Negative	27 (35.1%)	22 (47.8%)	49 (39.8%)	
Not Determined	3 (3.9%)	1 (2.2%)	4 (3.3%)	
Missing	4	0	4	

^1^ Chi-Square *p*-value; ^2^ Fisher exact *p*-value.

**Table 3 antibiotics-12-01541-t003:** Participants’ characteristics by birthplace and methicillin sensitivity of *S. aureus* and genotype USA300.

	US-Born					Non-US-Born				
	MRSA USA300 (*n* = 30)	MRSA Non-USA300 (*n* = 17)	MSSA USA300 (*n* = 8)	MSSA Non-USA300 (*n* = 26)	*p*-Value	MRSA USA300 (*n* = 9)	MRSA Non-USA300 (*n* = 11)	MSSA USA300 (*n* = 4)	MSSA Non-USA300 (*n* = 22)	*p*-Value
**Gender,** ***n* (%)**					0.47 ^1^					0.89 ^2^
Female	14 (46.7%)	9 (52.9%)	3 (37.5%)	8 (30.8%)		4 (44.4%)	5 (45.5%)	1 (25.0%)	8 (36.4%)	
Male	16 (53.3%)	8 (47.1%)	5 (62.5%)	18 (69.2%)		5 (55.6%)	6 (54.5%)	3 (75.0%)	14 (63.6%)	
**Race,** ***n* (%)**					0.98 ^2^					0.41 ^2^
Black	10 (50.0%)	5 (38.5%)	3 (50.0%)	9 (45.0%)		0 (0.0%)	0 (0.0%)	0 (0.0%)	3 (25.0%)	
White	5 (25.0%)	5 (38.5%)	2 (33.3%)	5 (25.0%)		2 (50.0%)	0 (0.0%)	0 (0.0%)	4 (33.3%)	
Other Race	5 (25.0%)	3 (23.1%)	1 (16.7%)	6 (30.0%)		2 (50.0%)	4 (100.0%)	2 (100.0%)	5 (41.7%)	
Missing	10	4	2	6		5	7	2	10	
**Ethnicity,** ***n* (%)**					0.87 ^1^					0.19 ^2^
Hispanic	17 (60.7%)	10 (58.8%)	5 (62.5%)	11 (50.0%)		6 (66.7%)	10 (100.0%)	4 (100.0%)	13 (76.5%)	
Non-Hispanic	11 (39.3%)	7 (41.2%)	3 (37.5%)	11 (50.0%)		3 (33.3%)	0 (0.0%)	0 (0.0%)	4 (23.5%)	
Missing	2	0	0	4		0	1	0	5	
**Age (years), *n* (%)**					0.46 ^2^					0.08 ^2^
<19	2 (6.7%)	4 (23.5%)	1 (12.5%)	3 (11.5%)		0 (0.0%)	0 (0.0%)	0 (0.0%)	0 (0.0%)	
19–45	20 (66.7%)	8 (47.1%)	2 (25.0%)	13 (50.0%)		5 (55.6%)	10 (90.9%)	4 (100.0%)	12 (54.5%)	
45–65	7 (23.3%)	4 (23.5%)	4 (50.0%)	8 (30.8%)		4 (44.4%)	1 (9.1%)	0 (0.0%)	10 (45.5%)	
>65	1 (3.3%)	1 (5.9%)	1 (12.5%)	2 (7.7%)		0 (0.0%)	0 (0.0%)	0 (0.0%)	0 (0.0%)	
**Marital Status,** ***n* (%)**					0.50 ^2^					0.85 ^2^
Couple	6 (20.0%)	5 (29.4%)	2 (25.0%)	3 (11.5%)		5 (62.5%)	8 (72.7%)	3 (75.0%)	12 (57.1%)	
Single	24 (80.0%)	12 (70.6%)	6 (75.0%)	23 (88.5%)		3 (37.5%)	3 (27.3%)	1 (25.0%)	9 (42.9%)	
Missing	0	0	0	0		1	0	0	1	
**Education,** ***n* (%)**					0.71 ^2^					0.77 ^2^
High School or lower	19 (63.3%)	11 (64.7%)	5 (62.5%)	19 (76.0%)		5 (62.5%)	6 (54.5%)	4 (100.0%)	15 (68.2%)	
College	10 (33.3%)	4 (23.5%)	2 (25.0%)	5 (20.0%)		0 (0.0%)	1 (9.1%)	0 (0.0%)	2 (9.1%)	
Bachelor or higher	1 (3.3%)	2 (11.8%)	1 (12.5%)	1 (4.0%)		3 (37.5%)	4 (36.4%)	0 (0.0%)	5 (22.7%)	
Missing	0	0	0	1		1	0	0	0	
**Health Insurance, *n* (%)**					0.79 ^2^					0.86 ^2^
Private/Other	8 (26.7%)	2 (11.8%)	1 (12.5%)	3 (11.5%)		2 (25.0%)	2 (18.2%)	1 (25.0%)	5 (23.8%)	
Public (Medicare or Medicaid)	20 (66.7%)	13 (76.5%)	6 (75.0%)	20 (76.9%)		2 (25.0%)	4 (36.4%)	0 (0.0%)	8 (38.1%)	
None	2 (6.7%)	2 (11.8%)	1 (12.5%)	3 (11.5%)		4 (50.0%)	5 (45.5%)	3 (75.0%)	8 (38.1%)	
Missing	0	0	0	0		1	0	0	1	
**Health Quality,** ***n* (%)**					0.31 ^2^					0.68 ^2^
Good	17 (58.6%)	11 (73.3%)	5 (62.5%)	17 (68.0%)		7 (87.5%)	9 (81.8%)	2 (50.0%)	14 (63.6%)	
Fair	10 (34.5%)	3 (20.0%)	1 (12.5%)	8 (32.0%)		1 (12.5%)	2 (18.2%)	2 (50.0%)	6 (27.3%)	
Poor	2 (6.9%)	1 (6.7%)	2 (25.0%)	0 (0.0%)		0 (0.0%)	0 (0.0%)	0 (0.0%)	2 (9.1%)	
Missing	1	2	0	1		1	0	0	0	
**Income, *n* (%)**					0.35 ^2^					0.88 ^2^
<USD 40,000	25 (96.2%)	10 (83.3%)	4 (80.0%)	17 (89.5%)		5 (83.3%)	5 (83.3%)	2 (66.7%)	15 (88.2%)	
≥USD 40,000	1 (3.8%)	2 (16.7%)	1 (20.0%)	2 (10.5%)		1 (16.7%)	1 (16.7%)	1 (33.3%)	2 (11.8%)	
Missing	4	5	3	7		3	5	1	5	
**Years in the US,** **Average** **(SD)**	36.1 (13.2)	31.9 (17.9)	42.4 (15.1)	33.8 (16.8)	0.44 ^3^	11.7 (6.0)	7.8 (8.3)	7.3 (1.2)	17.8 (11.9)	0.08 ^3^
**First Time Infection, *n* (%)**					0.27 ^1^					0.42 ^2^
Yes	18 (62.1%)	13 (86.7)	7 (87.5%)	15 (62.5%)		7 (87.5%)	7 (63.6%)	3 (75.0%)	19 (86.4%)	
No	11 (37.9%)	2 (13.3%)	1 (12.5%)	9 (37.5%)		1 (12.5%)	4 (36.4%)	1 (25.0%)	3 (13.6%)	
Missing	1	2	0	2		1	0	0	0	
**Crowded Life Environment,** ***n* (%)**					0.08 ^1^					0.82 ^2^
Yes	20 (76.9%)	11 (73.3%)	5 (62.5%)	11 (44.0%)		7 (87.5%)	8 (72.7%)	3 (75.0%)	15 (68.2%)	
No	6 (23.1%)	4 (26.7%)	3 (37.5%)	14 (56.0%)		1 (12.5%)	3 (27.3%)	1 (25.0%)	7 (31.8%)	
Missing	4	2	9	1		1	0	0	0	
**Healthcare Exposure, *n* (%)**					0.84 ^1^					0.46 ^2^
Yes	9 (33.3%)	7 (46.7%)	3 (37.5%)	11 (42.3%)		3 (37.5%)	2 (18.2%)	0 (0.0%)	8 (36.4%)	
No	18 (66.7%)	8 (53.3%)	6 (62.5%)	15 (57.7%)		5 (62.5%)	9 (81.8%)	4 (100.0%)	14 (63.6%)	
Missing	3	2	0	0		1	0	0	0	
**Animal** **Contact,** ***n* (%)**					0.42 ^1^					0.06 ^2^
Yes	14 (50.0%)	6 (40.0%)	2 (25.0%)	8 (30.8%)		3 (37.5%)	3 (27.3%)	3 (75.0%)	3 (13.6%)	
No	14 (50.0%)	9 (60.0%)	6 (75.0%)	18 (69.2%)		5 (62.5%)	8 (72.7%)	1 (25.0%)	19 (86.4%)	
Missing	2	2	0	0		1	0	0	0	
**Previous Wounds,** ***n* (%)**					0.19 ^1^					0.80 ^2^
Yes	9 (31.0%)	4 (26.7%)	2 (25.0%)	14 (53.8%)		1 (12.5%)	2 (18.2%)	1 (25.0%)	7 (31.8%)	
No	20 (69.0%)	11 (73.3%)	6 (75.0%)	12 (46.2%)		7 (87.5%)	9 (81.8%)	3 (75.0%)	15 (68.2%)	
Missing	1	2	0	0		1	0	0	0	
**Social Network, *n* (%)**					0.14 ^2^					1.0 ^2^
Yes	6 (21.4%)	6 (42.9%)	0 (0.0%)	5 (19.2%)		1 (12.5%)	2 (18.2%)	1 (25.0%)	5 (22.7%)	
No	22 (78.6%)	8 (57.1%)	8 (100%)	21 (80.8%)		7 (87.5%)	9 (81.8%)	3 (75.0%)	17 (77.3%)	
Missing	2	3	0	0		1	0	0	0	
**Household Crowding,** ***n* (%)**					0.39 ^1^					0.55 ^2^
<2 People/Room	8 (30.8%)	7 (50.0%)	1 (14.3%)	9 (39.1%)		1 (16.7%)	1 (11.1%)	0 (0.0%)	6 (31.6%)	
>2 People/Room	18 (69.2%)	7 (50.0%)	6 (85.7%)	14 (60.9%)		5 (83.3%)	8 (88.9%)	4 (100.0%)	13 (68.4%)	
Missing	4	3	1	3		3	2	0	3	
**Personal Hygiene,** ***n* (%)**					0.38 ^2^					0.10 ^2^
Sharing	12 (46.2%)	10 (71.4%)	5 (71.4%)	14 (63.6%)		1 (16.7%)	3 (33.3%)	2 (50.0%)	13 (68.4%)	
Not Sharing	14 (53.8%)	4 (28.6%)	2 (28.6%)	8 (36.4%)		5 (83.3%)	6 (66.7%)	2 (50.0%)	6 (31.6%)	
Missing	4	3	1	4		3	2	0	3	
**Hand Washing,** ***n* (%)**					0.81 ^1^					0.12 ^2^
<10 times/day	18 (62.1%)	10 (66.7%)	6 (75.0%)	19 (73.1%)		7 (87.5%)	5 (45.5%)	3 (75.0%)	18 (81.8%)	
>10 times/day	11 (37.9%)	5 (33.3%)	32 (25.0%)	7 (26.9%)		1 (12.5%)	6 (54.5%)	1 (25.0%)	4 (18.2%)	
Missing	1	2	0	0		1	0	0	0	

^1^ Chi-Square *p*-value; ^2^ Fisher exact *p*-value, ^3^ ANOVA *p*-value.

**Table 4 antibiotics-12-01541-t004:** Molecular characteristics of wound cultures identified as *S. aureus*, stratified by birthplace and genotype.

	US-Born					Non-US-Born				
	MRSA USA300 (*n* = 30)	MRSA Non-USA300 (*n* = 17)	MSSA USA300 (*n* = 8)	MSSA Non-USA300 (*n* = 26)	*p*-Value	MRSA USA300 (*n* = 9)	MRSA Non-USA300 (*n* = 11)	MSSA USA300 (*n* = 4)	MSSA Non-USA300 (*n* = 22)	*p*-Value
***mecA* Gene** **from *S. aureus* Wound, *n* (%)**					<0.0001					<0.0001
Positive	30 (100.0%)	12 (75.0%)	0 (0.0%)	0 (0.0%)		9 (100.0%)	9 (81.8%)	0 (0.0%)	0 (0.0%)	
Negative	0 (0.0%)	4 (25.0%)	7 (100.0%)	21 (87.5%)		0 (0.0%)	2 (18.2%)	4 (100.0%)	21 (95.5%)	
ND	0 (0.0%)	0 (0.0%)	0 (0.0%)	3 (12.5%)		0 (0.0%)	0 (0.0%)	0 (0.0%)	1 (4.5%)	
Missing	0	1	1	2		0	0	0	0	
**ACME Gene from *S. aureus* Wound, *n* (%)**					<0.0001					<0.0001
Type I	26 (86.7%)	3 (18.8%)	2 (28.6%)	0 (0.0%)		7 (77.8%)	2 (18.2%)	0 (0.0%)	0 (0.0%)	
Type III	1 (3.3%)	0 (0.0%)	0 (0.0%)	0 (0.0%)		0 (0.0%)	0 (0.0%)	0 (0.0%)	0 (0.0%)	
Negative	3 (10.0%)	13 (81.3%)	5 (71.4%)	21 (87.5%)		2 (25.0%)	9 (81.8%)	4 (100.0%)	21 (95.5%)	
ND	0 (0.0%)	0 (0.0%)	0 (0.0%)	3 (12.5%)		0 (0.0%)	0 (0.0%)	0 (0.0%)	1 (4.5%)	
Missing	0	1	1	2		0	0	0	0	
**SCCmec Gene from *S. aureus* Wound, *n* (%)**					<0.0001					<0.0001
IVa	29 (96.7%)	6 (37.5%)	0 (0.0%)	0 (0.0%)		9 (100.0%)	8 (72.7%)	0 (0.0%)	0 (0.0%)	
IVb	0 (0.0%)	1 (6.3%)	0 (0.0%)	0 (0.0%)		0 (0.0%)	0 (0.0%)	0 (0.0%)	0 (0.0%)	
IVc	1 (3.3%)	1 (6.3%)	0 (0.0%)	0 (0.0%)		0 (0.0%)	1 (9.1%)	0 (0.0%)	0 (0.0%)	
IVg	0 (0.0%)	1 (6.3%)	0 (0.0%)	0 (0.0%)		0 (0.0%)	0 (0.0%)	0 (0.0%)	0 (0.0%)	
IVh	0 (0.0%)	1 (6.3%)	0 (0.0%)	0 (0.0%)		0 (0.0%)	0 (0.0%)	0 (0.0%)	0 (0.0%)	
Novel Type	0 (0.0%)	2 (12.5%)	0 (0.0%)	0 (0.0%)		0 (0.0%)	0 (0.0%)	0 (0.0%)	0 (0.0%)	
Negative	0 (0.0%)	0 (0.0%)	1 (14.3%)	0 (0.0%)		0 (0.0%)	1 (9.1%)	0 (0.0%)	0 (0.0%)	
Not Determined	0 (0.0%)	4 (25.0%)	6 (85.7%)	24 (100.0%)		0 (0.0%)	1 (9.1%)	4 (100.0%)	22 (100.0%)	
**PVL Gene from *S. aureus* Wound, *n* (%)**					<0.0001					0.08
Positive	27 (90.0%)	9 (56.3%)	5 (71.4%)	6 (25.0%)		8 (88.9%)	6 (54.5%)	1 (25.0%)	8 (36.4%)	
Negative	3 (10.0%)	7 (43.8%)	2 (28.6%)	15 (62.5%)		1 (11.1%)	5 (45.5%)	3 (75.0%)	13 (59.1%)	
ND	0 (0.0%)	0 (0.0%)	0 (0.0%)	3 (12.5%)		0 (0.0%)	0 (0.0%)	0 (0.0%)	1 (4.5%)	
Missing	0	1	1	2		0	0	0	0	

**Table 5 antibiotics-12-01541-t005:** Multi-level regression analyses for *Staphylococcus aureus* USA300 risks.

	Unadjusted			Adjusted		
	OR	95% CI	*p*-Value	OR	95% CI	*p*-Value
**Birthplace**			0.06			0.03
US-born	2.55	0.99–7.20		3.20	1.15–9.89	
Non-US-born	ref			ref		
**Crowded Living**			0.16			0.07
≥2 People/Room	1.89	0.79–4.75		3.04	0.93–10.95	
<2 People/Room	ref			ref		
**Contact Animal**			0.03			0.10
Yes	2.53	1.08–6.06		2.13	0.87–5.31	
No	ref			ref		
**Age, Unit = 1 Year**	1.00	0.98–1.03	0.88	1.03	0.99–1.07	0.14

## Data Availability

De-identified data may be available upon written request to the Principal Investigator (jntobin@cdnetwork.org) and execution of a data usage agreement.

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
