# Peer review of "Molecular Epidemiologic and Geo-Spatial Characterization of Staphylococcus aureus Cultured from Skin and Soft Tissue Infections from United States-Born and Immigrant Patients Living in New York City"

_antibiotics, 2023, doi:10.3390/antibiotics12101541_

Round 1

Reviewer 1 Report

The publication has a high substantive value. There is no doubt about the methodology. The results of the research were presented clearly. The discussion was conducted clearly. However, the authors of this publication could refer to more studies on the same subject by other researchers in other parts of the world.  

Reviewer 2 Report

General comments:

In the manuscript entitled "Molecular Epidemiologic and Geo-spatial Characterization of Staphylococcus aureus Cultured from Skin and Soft Tissue Infections from U.S. Born and Immigrant Patients Living in New York City", the authors reveal that non-US born patients have unique risk factors and S. aureus molecular characteristics by analyzing many parameters. Although the manuscript contains interesting findings, some issues must be clarified.

Major specific comments:

1.          In the Abstract section, the research question or hypothesis needs to be clarified.

2.          In L. 331, the finding that frequent exposure to animals was a risk factor for a USA300 SSTI is quite interesting. Can the S. aureus USA300 strain colonize animals? How about other S. aureus strains? Why is this finding observed only for a USA300 SSTI? The authors are advised to discuss this point more.

3.          In L. 387, the authors argue that it is crucial to know the place of birth of patients (at a continental or country level in this study) for accurate S. aureus surveillance. However, as shown in Fig. 4, the distribution of S. aureus strains varies even within New York City, suggesting that a quite exact regional understanding is necessary for the strain estimation. The authors' argument and results seem controversial. The authors need to provide additional explanations on this point.

4.          I understand that the findings revealed in this study must be informative for clinicians, patients, and researchers in the US. However, what are the messages to readers in other countries? The authors are advised to provide some generalized arguments worldwide since the journal Antibiotics is an international journal.

Minor specific comments:

5.          In L. 90, the authors' hypothesis seems that the duration spent in the US could correlate with an increased risk for MRSA USA300 infections. If so, was the increased identification frequency of MRAS USA300 observed, depending on the duration spent in the US?

6.          In Tables 2 and 3, percentages are not provided in some parameters.

7.          In Fig. 4, what do red and yellow spots mean?

8.          In Fig. 4, although six community health centres were included in this study, Fig. 4 shows only five hospitals.

9.          The authors are advised to clarify if the racial composition in this study is consistent with the one of NYC since the proportion of Asians seems low in this study.

Reviewer 3 Report

The manuscript discusses the factors associated with S.aureus skin and soft tissue infections (SSTIs). They analyzed various risk factors and molecular characteristics of these infections among two major group of patients, those born in US and immigrants in New York city. Their study indicates a difference in infection prevalence based on birthplace i.e., those born in USA are more prone to the MRSA USA300 strain. While other factors such as gender, hygiene habits and animal exposure are other significant risk factors. Overall the work provides comprehensive and valuable insights that could inform more effective strategies for preventing and managing S.aureus infections. However, I worry about the long term implications of the identified risk factors or strains as things change over the time.

Major comments

1.      The sample collection was done in 2015 -2017, which is almost a decade old now. Does study still have relevance when it comes to the various factors they have studied. The data might be outdated or not reflective of the current situation. Since post covid a lot of things have changed in the health care system and other risk factors.

2.      The reliance on participant recall of certain risk factors such as sharing hygiene products and animal exposure can introduce a recall bias.

Minor editing 

Round 2

Reviewer 2 Report

The authors have addressed all the comments raised by this reviewer.